# Attitude towards Vaccination among Health Science Students before the COVID-19 Pandemic

**DOI:** 10.3390/vaccines9060644

**Published:** 2021-06-12

**Authors:** Pérez-Rivas Francisco Javier, Del Gallego-Lastra Ramón, Esteban-Garcimartín Ana, Marques-Vieira Cristina Maria Alves, Ajejas Bazán María Julia

**Affiliations:** 1Departamento de Enfermería, Facultad de Enfermería, Fisioterapia y Podología, Universidad Complutense de Madrid, Plaza Ramón y Cajal n° 3, Ciudad Universitaria, 28040 Madrid, Spain; rgallego@ucm.es (D.G.-L.R.); majejas@ucm.es (A.B.M.J.); 2Estudiante de Grado de Enfermería, Facultad de Enfermería, Fisioterapia y Podología, Universidad Complutense de Madrid, 28040 Madrid, Spain; aneste01@ucm.es; 3Center Interdisciplinary Research in Health, Institute of Health Sciences, Nursing School (Lisbon), Universidade Católica Portuguesa, 1649-023 Lisbon, Portugal; cristina_marques@ucp.pt; 4Academia Central de la Defensa, Escuela Militar de Sanidad, Ministerio de Defensa, 28040 Madrid, Spain

**Keywords:** attitudes, beliefs, health occupations, influenza vaccination, students, vaccination

## Abstract

Health science students are tomorrow’s health professionals, the duties of whom could include vaccination. This work examines the general attitude towards vaccination in students attending the Faculty of Nursing, Physiotherapy and Chiropody at a university in Madrid, Spain, using the ‘Attitudes and Behaviour With Regard To Vaccination Among Health Science Students Questionnaire’. The results were subjected to multivariate analysis to identify the influence of sex, the degree being pursued, and ‘course year’. The number of students vaccinated against influenza in the campaign preceding the present study was also recorded, as were the factors that influenced decision-making in this regard. A total of 934 students completed the questionnaire. Their beliefs regarding vaccination were positive (mean score 3.23 points out of 4), as was their behaviour (3.35/4). Their general attitude (all variables taken together) was therefore also good (3.27/4). Only 26.8% of the students had been vaccinated against influenza. Beliefs scores among the students of nursing in their more senior course years were significantly better than those recorded for all other groups. These students also showed the best general attitude towards vaccination and formed the largest group vaccinated against influenza. The results obtained are encouraging since nursing students are the most likely of future healthcare professionals to be involved in vaccination programmes.

## 1. Introduction

Vaccination is one of the most efficient public health interventions for controlling transmissible disease. Over recent decades systematic vaccination programmes have drastically reduced the mortality and morbidity associated with infectious disease [1], as well as the healthcare costs they incur [2].

Even though vaccination might be considered the all-time most efficient means of preventing disease [3] (with the exception of the provision of clean drinking water), and while vaccination coverage is very high in Europe, including Spain [4], recent years have seen the rise of anti-vaxxer movements which have led to an increasing rejection of, or at least hesitancy towards, vaccination [5,6].

The World Health Organisation’s Strategic Advisory Group of Experts on Immunisation defines vaccine hesitancy as “a delay in accepting or rejecting safe vaccines, despite the availability of vaccination services” [7]. This hesitancy is influenced by factors such as complacency, convenience, and confidence. While complacency and convenience are related to the perception of risk of disease and accessibility to vaccination services, confidence is defined as trust in the safety and effectiveness of vaccines and the health system that delivers them [6].

The idea that vaccines are not safe, that they contain dangerous adjuvants, or that they might have unknown long-term adverse effects has contributed towards vaccine hesitancy. Such beliefs lead to people overestimating the risks and underestimating the benefits of vaccination [8]. Unfortunately, they are not limited to the general population; certain healthcare professionals also hold them [9]. This alarming situation has led the WHO to declare vaccine hesitancy as one of the top 10 threats to world health [10].

The role played by healthcare professionals with respect to vaccine hesitancy is crucial [11,12]. Healthcare workers are usually the main source of information for patients regarding health matters, including vaccination, and usually enjoy their trust [13,14]. The attitude of the former towards recommending vaccination is therefore vital. The 2007 Summit of Independent European Vaccination Experts (SIEVE) [15] made it clear that strategies aimed at optimising vaccination coverage among adults and children in Europe ought to be directed towards healthcare professionals, focusing on their attitude towards vaccination since this could determine the vaccination-related decisions made by families under their care [9]. Despite these recommendations, however, several studies have shown that many such professionals are worried about the safety of vaccines [16,17], while others may find it difficult to respond to hesitant patients’ questions because they themselves are hesitant [18,19]. These findings are worrying since such negative attitudes could influence the decisions taken by patients regarding vaccination [11,14,16,20,21].

Health science students are tomorrow’s healthcare professionals, and the responsibility of making vaccination programmes effective will eventually fall to them. Knowing their general attitude towards vaccination is therefore important, as is implementing any intervention required to modify their attitude should this be necessary. The aims of the present work were (1) to determine their beliefs regarding vaccination, their behaviour regarding vaccination (in terms of auto-recommendation of vaccination and recommending it to others, etc.), and their general attitude (all variables scores taken as a whole) with regard to vaccination, (2) to determine whether any differences exist between students of nursing, chiropody and physiotherapy in these respects, (3) to determine whether scores for these variables change as their courses progress, and (4) to identify whether any of these variables affect their seeking vaccination against influenza.

## 2. Materials and Methods

This cross-sectional study involved 934 students, all attending the Faculty of Nursing Physiotherapy and Chiropody, Universidad Complutense de Madrid (Spain), who responded to the ‘Attitudes and Behaviour With Regard To Vaccination Among Health Science Students Questionnaire’ (ACVECS according to its Spanish initials) [22]. Sample size requirement analysis showed 310 respondents were needed for a 5% error and 95% confidence limits. All responders gave their informed consent to be included.

The ACVECS questionnaire examines beliefs, behaviour and general attitude towards vaccination. It is composed of 24 items; the first 15 of which examine beliefs, while the last 9 examine behaviour. Taken together, these 24 items determine the “general attitude” towards vaccination. All items were answered on a five-point Likert scale from 0 = totally disagree, to 4 = totally agree. For items 1, 2, 7, 8, 15 and 23, the scores have to be inverted before analysis given the way in which these questions are phrased.

The researchers contacted the coordinators of the Nursing, Physiotherapy and Chiropody degree courses in order to organise a meeting with the students present on different days between 15–30 January 2020. During this meeting the study was explained, and it was made clear that anonymity was guaranteed at all times (names were neither required nor requested). It was hoped that this would also safeguard against bias in the students’ answers caused through a desire to please their peers or teachers.

The students who wished to complete the questionnaire did so in situ over a period of 10–15 min, using their smartphones to access the questionnaire via the University’s virtual campus. It they were unable to log in they were allowed to complete the questionnaire later from another physical location. The questionnaire was completed using an on-line Google Forms^®^ form, which also collected information on respondent age, sex, degree being pursued, course year, and on whether the respondent had been vaccinated against influenza in the campaign preceding the study.

All information collected was transferred to a database designed using Microsoft Office Excel 2016. Beliefs, behaviour and general attitude scores of ≥3 were considered positive (i.e., vaccination-favourable), ≤1 negative and =2 considered neutral or indifferent.

The examined variables were first subjected to simple statistical description. The Kolmogorov-Smirnov test was used to examine the normality of distribution of the results. Quantitative variables were described as means ± standard deviation (SD) (normally distributed results) or median and interquartile range (not normally distributed). The Chi squared test was used to compare qualitative variables, and the Student *t* test or ANOVA to compare quantitative variables. Significance was set at *p* < 0.05; 95% confidence intervals (95%CI) were also determined. The effect of sex, age, the degree being pursued, and course year (1st, 2nd, 3rd or 4th) on beliefs, behaviour and general attitude was examined by regression analysis. In addition, associations were also sought between all the studied variables and whether the students had been vaccinated against influenza in the campaign preceding the study period. Those variables that showed a significant relationship in bivariate analysis, or that were considered important even though they showed no association, were included in binary logistic regression and multiple linear regression analyses. The relationships between dependent and independent variables in binary logistic regression analyses were determined via the *p* value and 95%CI of the exponents of the B coefficient (e^B^ = OR).

All analyses were undertaken using the Statistical Package for the Social Sciences (SPSS) v.25 for Windows (IBM©).

The study was approved by the Research Committee of the Faculty of Nursing, Physiotherapy and Chiropody, Universidad Complutense de Madrid, and by the Ethics in Research Committee of the Hospital Universitario Clínico San Carlos. All work was performed in adherence to the principles of the Declaration of Helsinki (2013 version). All data were treated adhering to General Data Protection Regulation (GDPR) 2016/679, 27th April 2016, and the Spanish Ley orgánica de protección de datos y garantía de derechos digitales (LOPDGDD; Data Protection And Guarantee Of Digital Rights) 3/2018, 5 December.

## 3. Results

A total of 934 students responded to the questionnaire. This figure represents 59.1% of the entire student body attending the faculty (65.6% of whom are nursing students, 16.6% physiotherapy students, and 17.8% chiropody students) (Table 1). Of those who responded, 31.2% were in their first year, 32.1% were in their second, 19.2% in their third, and 17.5% in their fourth. At least 30 students from each degree and course year enrolled, except for the third year of chiropody (*n* = 28). In this case, 79.5% of the responders were female. 66.9% of the respondents were students of nursing, 17.3% of physiotherapy and 15.8% of chiropody, reflecting well the distribution for the faculty as a whole (Table 2). The mean age of the respondents was 21.3 ± 0.34 years; the mean age of the chiropody students (22.5 ± 0.4 years) was a little higher than that of the nursing (21.1 ± 5.4 years) and physiotherapy students (20.6 ± 0.24 years) (*p* < 0.01).

As a whole, the responding students scored a mean 3.23 (out of 4) in terms of belief, 3.35 in behaviour, and 3.27 in general attitude. Table 3 shows the distribution of these three dimensions with respect to sex, the degree being pursued, and course year. Female students obtained better beliefs, behaviour and general attitude scores than their male colleagues (*p* < 0.05). The students of nursing returned the highest scores for all three dimensions (*p* < 0.01). Moreover, their belief and general attitude scores significantly improved year on year as they progressed through their course (*p* < 0.05). The belief scores for the physiotherapy students improved, but not homogenously year on year.

For 20 of the 24 (83%) items in the questionnaire, over 70% of the students gave a positive response. For example, only 0.9% would not recommend the established vaccination schedule to their patients, and just 0.6% would not give their patients all the information on the effectiveness and possible adverse effects of a vaccine. However, 15.6% indicated that they did not consider it their ethical duty to be vaccinated against influenza, and 17.2% would not be vaccinated every year. Overall, the nursing students were those with the most positive beliefs and behaviour scores, followed by the physiotherapy and chiropody students (Table 4).

Multivariate analysis revealed female students, and to be following a nursing degree to be related to a more positive behaviour. Following a nursing degree and to be in more senior years of that degree was associated with a more positive score for beliefs. Female students, following a nursing degree, and being in the more senior years of that degree were also related to a better general attitude towards vaccination (Table 5).

In this case, 250 students (26.8%) declared having been vaccinated against influenza in the campaign preceding the study; no significant difference was seen between male (25.2%) and female (27.2%) students. In total, 32.5% of the nursing students were vaccinated, 26.8% of the chiropody students, and 17.9% of the physiotherapy students (*p* < 0.001). In terms of course year, second year students were those who most often sought vaccination (40.3%), followed by third year (31.5%), fourth year (21.3%), and first year (13.0%) students (*p* < 0.001). Those students who scored positive behaviour scores sought vaccination more often than those with poorer scores (28.7% vs. 13.4%; *p* < 0.001).

The likelihood of seeking vaccination increased with age (OR 1.04 [1.01–1.07]). The nursing students were more likely to have sought vaccination to influenza than either the physiotherapy (OR 2.61 [1.30–5.21]) or chiropody (OR 4.53 [2.57–7.96] students. Those students with positive belief scores were also more likely to have sought vaccination than those with poorer scores (OR 2.06 [1.18–3.60]); the same was seen for those students with positive behaviour scores (OR 3.74 [2.17–6.44]) (Table 6).

## 4. Discussion

A total of 934 students completed the questionnaire—59.1% of the faculty’s entire student body. This is a high percentage compared to similar studies [22,23,24,25].

In general, the present students had a good general attitude towards vaccination. Good scores were returned for all the dimensions examined; indeed, they were higher than those obtained in the study in which the questionnaire was originally validated [22]. Studies similar to the present performed in Serbia [23], Florida (USA) [26] and Australia [27] obtained even more positive results, while others performed in the USA [25], Germany [28], Canada [24] and Italy [29] recorded fewer positive results.

In the present work, female students were associated with a more positive general attitude towards vaccination; this has also been reported from the USA [30], Poland [31] and a previous study undertaken in Spain [32]. In contrast, an Italian study found male nursing students to have a better attitude [33]. It is not clear why women have a more positive attitude towards vaccination than men.

The nursing students returned better beliefs, behaviour and general attitude scores than did the students pursuing the other degrees. This might be explained in that nurses have far more responsibility than physiotherapists or chiropodists viz a viz vaccine administration [34], and that their studies include much more extensive content on immunisation. Even though the present study did not examine student knowledge of vaccination, the literature contains ample evidence of a correlation between level of knowledge and attitude towards vaccination [23,24,26]. The better general attitude shown by the students of nursing compared to those of other professions could not, however, be confirmed by the literature; no studies were found assessing the attitudes of physiotherapy and chiropody students. Some studies show students of nursing to be less knowledgeable and to have a poorer attitude towards vaccination than students of medicine and pharmacy [24,25].

The general attitude of the nursing students became better year-on year as their course progressed. This has been reported previously in two Spanish studies [22,32] and other international studies [23,35]. This is probably due to their increased knowledge in their more senior years, reflecting the above-mentioned relationship between knowledge and attitude.

Even though both Spanish [36] and international bodies [37] recommend vaccination against influenza for students of health sciences, only 26.7% of the 934 respondents had been so vaccinated. This is lower than that reported from the USA [38] and Australia [39], similar to that recorded in Israel [40] and Ireland [41], and very much higher than that reported in studies from China [42], Italy [43], Poland [44] and indeed from another study performed in Spain [45].

In agreement with other studies, multivariate analysis showed that female students [45], increasing age [42,45], studying nursing rather than physiotherapy [39,44], and having more positive beliefs and behaviour [38,41,42] to be associated with seeking vaccination against influenza.

The low coverage of influenza vaccination among the chiropody students (12.2% compared to 32.5% among nursing students and 17.9% among physiotherapy students) might be influenced by the lesser content on vaccination in their course. In addition, the chiropody students do not undertake their practical training at external health centres but at the University’s chiropody clinic. Similarly, the finding that the second-year nursing students had the highest coverage might be explained by their training at external health centres at the time of the yearly influenza vaccination campaign.

It is important to note that the present results were not influenced by the COVID-19 pandemic since they were collected before it began in Spain. Even though news items on the disease were published at that time, the present high level of media attention had by no means been reached. It would be of interest to repeat this study when the pandemic is over.

The present work suffers the limitation that the study subjects were only those willing to take part who were present on the days when the project was explained. Even though those who responded represented 59.1% of the entire student body, no information was collected on those who did not participate. It remains possible that their characteristics were different to those who took part.

Finally, the present results were all obtained from health science students at one university; they may not be extrapolatable to those attending other universities.

## 5. Conclusions

The beliefs, behaviour and general attitude towards vaccination among the study subjects can be said to be positive, especially among the students of nursing, and they improve year on year as these students progress through their course. These findings are encouraging since it is largely nurses who are responsible for administering vaccines; they also reflect a positive influence of the instruction they receive. Even though physiotherapists and chiropodists are not at the forefront of vaccination campaigns, they are health professionals, and it might be useful to increase the content on vaccination in their courses.

The coverage of vaccination against influenza, while better than that seen in some studies and similar to that recorded in others, was still quite low. Debate is required on how to increase this coverage among health science students, especially among those undertaking training periods at health centres. The finding that those students with better beliefs and behaviour scores were those among whom this coverage was greatest reflects a need to design strategies that improve the general attitude towards vaccination of those students with lower scores.

## Figures and Tables

**Table 1 vaccines-09-00644-t001:** Participation in the study by degree being pursued and course year.

Degree	Total Matriculated	Total Participating Subjects	Participating Subjects per Year			
			1	2	3	4°
	*n* (%)	*n* (%)	*n* (%)	*n* (%)	*n* (%)	*n* (%)
**Nursing**	1039 (65.6)	624 (66.9)	220 (35.2)	192 (30.8)	120 (19.3)	92 (14.7)
**Physiotherapy**	262 (16.6)	162 (17.3)	40 (24.6)	50 (30.8)	30(18.5)	42 (25.9)
**Chiropody**	281 (17.8)	148 (15.8)	32 (21.6)	58 (39.2)	28 (18.9)	30 (20.3)
**TOTAL**	1582 (100.0)	934 (100.0)	292 (31.2)	300 (32.1)	178 (19.2)	164 (17.5)

**Table 2 vaccines-09-00644-t002:** Distribution of participating subjects by sex and course year in each degree pursued.

Variables		Total		Nursing		Physiotherapy		Chiropody		*p* Value
		*n* (%)	95%CI	*n* (%)	95%CI	*n* (%)	95%CI	*n* (%)	95%CI	
Sex	Female	743 (79.5)	(77.0–82.1)	532 (85.3)	(83.0–87.6)	101 (62.3)	(59.2–65.4)	110 (74.3)	(71.5–77.1)	*p* < 0.01
	Male	191 (20.5)	(17.9–23.2)	92 (14.7)	(12.4–17.0)	61 (37.7)	(34.6–40.8)	38 (25.7)	(22.9–28.5)	
Course year	1st	292 (31.2)	(28.2–34.2)	220 (35.3)	(32.2–38.4)	40 (24.7)	(21.9–27.5)	32 (21.6)	(19.0–24.2)	0.062
	2nd	300 (32.2)	(29.1–35.1)	192 (30.8)	(27.8–33.8)	50 (30,9)	(27.9–33.9)	58 (39.2)	(36.1–42.3)	
	3rd	178 (19.0)	(16.5–21.5)	120 (19.2)	(16.7–21.7)	30 (18.5)	(16.0–21.0)	28 (18.9)	(16.4–21.4)	
	4th	164 (17.6)	(15.1–19.9)	92 (14.7)	(12.4–17.0)	42 (25.9)	(23.1–28.7)	30 (20.3)	(17.7–22.9)	

**Table 3 vaccines-09-00644-t003:** Distribution of scores for ACVECS questionnaire dimensions with respect to sex, degree being pursued and course year. Madrid, Spain, 2020.

Variables	Categories	Beliefs Mean ± SD	*p* Value	Behaviour Mean ± SD	*p* Value	General Attitude Mean ± SD	*p* Value
Sex	Male	3.15 ± 0.46	0.02	3.24 ± 0.53	0.001	3.19 ± 0.48	0.005
	Female	3.24 ± 0.50		3.38 ± 0.50		3.29 ± 0.44	
Degree	Nursing	3.28 ± 0.43	*p* < 0.0001	3.41 ± 0.45	*p* < 0.0001	3.33 ± 0.40	*p* < 0.0001
	Physiotherapy	3.07 ± 0.51		3.16 ± 0.61		3.10 ± 0.51	
	Chiropody	3.14 ± 0.53		3.29 ± 0.35		3.19 ± 0.52	
Nursing course year	1st	3.19 ± 0.45	*p* < 0.0001	3.40 ± 0.46	0.125	3.27 ± 0.43	0.005
	2nd	3.29 ± 0.42		3.43 ± 0.43		3.34 ± 0.38	
	3rd	3.32 ± 0.39		3.35 ± 0.48		3.33 ± 0.39	
	4th	3.42 ± 0.42		3.50 ± 0.41		3.45 ± 0.39	
Physiotherapy course year	1st	3.07 ± 0.42	0.015	3.25 ± 0.59	0.356	3.14 ± 0.46	0.070
	2nd	3.02 ± 0.48		3.10 ± 0.60		3.05 ± 0.49	
	3rd	2.88 ± 0.64		3.02 ± 0.79		2.94 ± 0.66	
	4th	3.26 ± 0.45		3.23 ± 0.48		3.25 ± 0.43	
Chiropody course year	1st	3.13 ± 0.66	0.947	3.38 ± 0.60	0.767	3.22 ± 0.62	0.964
	2nd	3.13 ± 0.48		3.25 ± 0.55		3.17 ± 0.48	
	3rd	3.20 ± 0.53		3.26 ± 0.63		3.22 ± 0.55	
	4th	3.12 ± 0.50		3.29 ± 0.50		3.18 ± 0.45	

**Table 4 vaccines-09-00644-t004:** ACVECS questionnaire: frequencies and percentages of students showing negative scores with respect to degree.

Questionnaire Items	Total *n* (%)	Degree			
		Nursing *n* (%)	Physiotherapy *n* (%)	Chiropody *n* (%)	*p* Value
1.- I have doubts about the effectiveness of vaccines	104 (11.1)	78 (12.5)	12 (10.5)	14 (9.5)	0.341
2.- I would rather have influenza than be vaccinated against it	96 (10.3)	61 (9.8)	16 (9.9)	19 (12.8)	0.736
3.- I am convinced that marketed vaccines are safe	73 (7.8)	44 (7.1)	19 (11.7)	10 (6.8)	0.067
4.- I am interested in learning more about vaccination	71 (7.6)	31 (5.0)	17.3 (12)	12 (8.1)	*p* < 0.001
5.- I believe it important to check my vaccination status before travelling to a tropical country such as Mexico or Thailand	7 (0.7)	4 (0.6)	1 (0.6)	2 (1.4)	0.210
6.- National and international vaccine campaigns are cost-effective	146 (15.6)	103 (16.5)	22 (13.6)	21 (11.2)	0.296
7.- It is not worth being vaccinated against a disease for which effective treatment exists	56 (6.0)	33 (5.3)	16 (9.9)	7 (4.7)	0.006
8.- Vaccinating the adult population is not important	30 (3.2)	15 (2.4)	9 (5.6)	6 (4.1)	0.106
9.- Health science students are ethically obliged to be vaccinated against influenza	146 (15.6)	88 (14.1)	34 (21.0)	24 (16.2)	0.147
10.- Being vaccinated myself has a positive influence on the behaviour of my patients	51 (5.5)	30 (4.8)	12 (7.4)	9 (6.1)	*p* < 0.001
11.- Students should be vaccinated to reduce the transmission of infectious diseases in hospitals	22 (2.4)	10 (1.6)	5 (3.1)	7 (4.7)	0.043
12.- I should review my vaccination status before starting clinical training	28 (3.0)	20 (3.2)	1 (0.6)	7 (4.7)	0.488
13.- I should be vaccinated against influenza every year, even it means missing hours of practical training	159 (17.0)	81 (13.0)	45 (27.8)	33 (22.3)	*p* < 0.001
14.- I would be vaccinated irrespective of what my peers might do	25 (2.7)	11 (1.8)	5 (3.1)	9 (6.1)	0.018
15.- If I am in good health there is no need to be vaccinated	55 (5.9)	33 (5.3)	12 (7.4)	10 (6.8)	0.002
16.- I would recommend my patients adhere to the established vaccination calendar	8 (0.9)	3 (0.5)	3 (1.9)	2 (1.4)	*p* < 0.001
17.- I would inform my patients of the effectiveness, indications and side effects of each vaccine	6 (0.6)	1 (0.2)	4 (2.5)	1 (0.7)	*p* < 0.001
18.- I would travel to a tropical country only after consulting Spain’s International Vaccination about the vaccines I require	35 (3.7)	22 (3.5)	11 (6.8)	2 (1.4)	0.050
19.- I would be vaccinated against HIV when a vaccine becomes available and when shown to be acceptably safe and effective	32 (3.4)	24 (3.8)	6 (3.7)	2 (1.4)	0.190
20.- If being vaccinated against influenza were readily accessible to me I would be vaccinated every year	64 (6.9)	36 (5.8)	16 (9.9)	12 (8.1)	0.083
21.- I would be vaccinated against anything my doctor recommends, even if I have to pay for it	87 (9.3)	56 (9.0)	21 (13.0)	10 (6.8)	0.006
22.- When I begin work at a hospital I will make sure I am vaccinated against everything preventable	10 (1.1)	3 (0.5)	3 (1.9)	4 (2.7)	0.029
23.- I would only be vaccinated in exceptional circumstances (epidemics, health alerts etc.	112 (12.0)	70 (11.2)	25 (15.4)	17 (11.5)	0.333
24.- I will be vaccinated against influenza every year I have clinical training	122 (13.1)	67 (10.7)	21 (13.0)	34 (23.0)	*p* < 0.001

**Table 5 vaccines-09-00644-t005:** Multivariate analysis: relationships between beliefs, behaviour and general attitude and the variables sex, age degree being pursued and course year.

Variables	Beliefs (B Coefficient)	*p* Value	Behaviour (B Coefficient)	*p* Value	General Attitude (B Coefficient)	*p* Value
Sex	−0.07 (−0.14, 0.10)	0.088	−0.11 (−0.19, −0.03)	0.008	−0.08 (−0.16, −0.01)	0.026
Age	−0.003 (−0.009, 0.002)	0.267	−0.002 (−0.006, 0.006)	0.984	−0.002 (−0.008, 0.004)	0.469
Degree	−0.10 (−0.14, −0.05)	*p* < 0.001	−0.09 (−0.13, −0.04)	*p* < 0.001	−0.09 (−0.13, −0.05)	*p* < 0.001
Course year	0.06 (0.03, 0.09)	*p* < 0.001	0.002 (−0.030, 0.033)	0.916	0.04 (0.01–0.07)	0.009

**Table 6 vaccines-09-00644-t006:** Bivariate and multivariate analyses: effect of sex, age, degree being pursued, course year, beliefs and behaviour on seeking vaccination to influenza.

Variables	β	OR (95CI%)	*p* Value	β	OR (95CI%)	*p* Value
**Sex**						
females vs. males	0.10	1.11 (0.77–1.60)	0.567			
**Age**	0.03	1.03 (1.00–1.05)	0.029	0.04	1.04(1.01–1.07)	0.006
**Degree**						
nursing vs. physiotherapy	0.79	2.21 (1.43–3.42)	*p* < 0.001	0.96	2.61(1.30–5.21)	0.007
nursing vs. chiropody	1.25	3.48 (2.07–5.86)	*p* < 0.001	1.51	4.53(2.57–7.96)	*p* < 0.001
**Course year**						
2nd vs. 1st	1.51	4.52 (2.99–6.82)	*p* < 0.001	1.82	6.20(3.96–9.71)	*p* < 0.001
3rd vs. 1st	1.12	3.07 (1.93–4.89)	*p* < 0.001	1.30	3.65(2.21–6.03)	*p* < 0.001
4th vs. 1st	0.60	1.81 (1.09–3.01)	*p* = 0.021	0.52	1.68(0.96–2.91)	*p* = 0.067
**Beliefs**	1.55	4.73 (3.2–6.99)	*p* < 0.001	0.72	2.06(1.18–3.60)	0.011
**Behaviour**	1.59	4.89 (3.32–7.22)	*p* < 0.001	1.32	3.74(2.17–6.44)	*p* < 0.001
**General attitude**	1.85	6.33 (4.11–9.76)	*p* < 0.001

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
