# Peer review of "Attitude towards Vaccination among Health Science Students before the COVID-19 Pandemic"

_vaccines, 2021, doi:10.3390/vaccines9060644_

Round 1

Reviewer 1 Report

  1. Ln17 Abstract - "these students" is not appropriate here, since health care students also include other branches that are not included in the study.
  2. Ln 93 – if table 4 represents the 24 questions authors asked in questionnaire, please mention this here. If they don’t represent, then Ln 96 has no meaning – “items 1, 2, 7, 8, 15 and 23”, since readers don’t understand what these questions represent. Please clarify. 
  3. Ln 112-113 – what is the score between 1 and 3 represented?
  4. The sentence (Ln 170 onwards) suggests “Overall, the nursing students were those with the most positive beliefs and behaviour scores, followed by the physiotherapy and chiropody students (Table 4).”, while table title suggests its negative attitude that is scored. For e.g. As per title of table 4, is it that 44 nursing students (First row, first column) that answered “I am (NOT?) convinced that marketed vaccines are safe”? Please explain clearly.
  5. In the same context as above clarification I sought, did 78 nursing students answered negatively (or in theory had a good opinion) about the question “I have doubts about the effectiveness of vaccines”?
  6. Please explain the parameter (CI?) presented in parenthesis in table 5. What is meant by (B)?
  7. Please narrate/expand the parameters presented in formula (ln 129); Is it odd’s ratio (OR) that was calculated and presented in Table 6?
    Authors should pay attention to proper editing and follow the journal article pattern – e.g. Abstract has ‘method’, ‘result’ etc written without proper highlight, which should be displayed with appropriate discerning patterns. English should be improved – e.g. “The number of students who were vaccinated against influenza in the campaign” gives an impression that the present work is a campaign to vaccinate (ln 21-22); ‘other type of student’ (ln 31); ‘most likely of future’ (ln 33); should use a better way to represent ‘female sex’ (re-assigned ln 1, 3 page 11, immediately after the table 4) and ln 33, page 12 – is ‘female students’ better than ‘female sex’? there is a repetition of three ‘to’s in that sentence Spelling/grammatical errors - Ln 73 “vasccination”; ln 10, page 11 – 32.5% ‘of’ Other minor corrections needed – Ln 135 grey highlight between “All data”; colored column in table 6; It is better if a native English speaker goes through the article narratives.

Author Response

Response to Reviewer 1 Comments

Point 1: Ln17 Abstract - "these students" is not appropriate here, since health care students also include other branches that are not included in the study.

The text is modified, specifying that they are students of nursing, physiotherapy and podiatry.

Point 2: Ln 93 – if table 4 represents the 24 questions authors asked in questionnaire, please mention this here. If they don’t represent, then Ln 96 has no meaning – “items 1, 2, 7, 8, 15 and 23”, since readers don’t understand what these questions represent. Please clarify.

Indeed, Table 4 represents the 24 questions authors asked in questionnaire. A clarification is included in Table 4, specifying that the questions appearing in that table are those of the ACVECS questionnaire ('Attitudes and Behaviour With Regard To Vaccination Among Health Science Students Questionnaire').

Point 3: Ln 112-113 – what is the score between 1 and 3 represented?

Add the aclarification:

“Beliefs, behaviour and general attitude scores of ≥3 were considered positive (i.e., vaccination-favourable), ≤1 considered negative and =2 considered neutral or indifferent”

Point 4: The sentence (Ln 170 onwards) suggests “Overall, the nursing students were those with the most positive beliefs and behaviour scores, followed by the physiotherapy and chiropody students (Table 4).”, while table title suggests its negative attitude that is scored. For e.g. As per title of table 4, is it that 44 nursing students (First row, first column) that answered “I am (NOT?) convinced that marketed vaccines are safe”? Please explain clearly.

In the same context as above clarification I sought, did 78 nursing students answered negatively (or in theory had a good opinion) about the question “I have doubts about the effectiveness of vaccines”?

As noted in the title of the table, the data shown in the Table are the frequency and percentage of students who responded negatively to each of the 24 questions in the questionnaire (score ≤ 1 on the likert scale).

The wording of the sentence indicated by the reviewer is changed to make it clearer what we are referring to:

“Overall, nursing students scored the least negatively on beliefs and behaviour, followed by physiotherapy and podiatry students (Table 4)"

Point 5: What is meant by (B)?

This is the Beta regression coefficient. It indicates the average change in the dependent variable for each unit of change in the independent variable. Explanation is added in the table.

Please explain the parameter (CI?) presented in parenthesis in table 5.

This is the confidence interval of the Beta regression coefficient. They report the limits between which we can expect the population value of each regression coefficient to lie. When it includes 0, the coefficient is not statistically significant.

Point 6: Please narrate/expand the parameters presented in formula (ln 129); Is it odd’s ratio (OR) that was calculated and presented in Table 6?

Indeed, these are the ORs shown in Table 6, which have been calculated considering the exponents of the Beta regression coefficients (eB=OR).

Point 7: Authors should pay attention to proper editing and follow the journal article pattern – e.g. Abstract has ‘method’, ‘result’ etc written without proper highlight, which should be displayed with appropriate discerning patterns.

The headings in the abstract have been removed and adjusted to the recommended 200 words.

Point 8: English should be improved – e.g. “The number of students who were vaccinated against influenza in the campaign” gives an impression that the present work is a campaign to vaccinate (ln 21-22); ‘other type of student’ (ln 31); ‘most likely of future’ (ln 33); should use a better way to represent ‘female sex’ (re-assigned ln 1, 3 page 11, immediately after the table 4) and ln 33, page 12 – is ‘female students’ better than ‘female sex’? there is a repetition of three ‘to’s in that sentence Spelling/grammatical errors - Ln 73 “vasccination”; ln 10, page 11 – 32.5% ‘of’ Other minor corrections needed – Ln 135 grey highlight between “All data”; colored column in table 6; It is better if a native English speaker goes through the article narratives.

The text has been sent to a native English translator who has been in charge of grammatical and spelling checking.

Punto 9: should use a better way to represent ‘female sex’ (re-assigned ln 1, 3 page 11, immediately after the table 4) and ln 33, page 12 – is ‘female students’ better than ‘female sex’?

Female sex' is replaced by 'female students'.

Punto 10: ln 10, page 11 – 32.5% ‘of’

Replace 'or' with 'of'.

Punto 11: Other minor corrections needed – Ln 135 grey highlight between “All data”;

Removed the grey background between the words 'All' and 'Data'.

Punto 12: colored column in table 6

The shading of the column in Table 6 is removed.

Reviewer 2 Report

The article addresses a topic of interest, especially at this time when vaccination appears continuously in the press and on TV, with an informational treatment that is usually superficial and without the necessary rigor.

The sample of participants is adequate for the proposed objectives.

Likewise, the instrument and the data analysis are also adequate to achieve the objectives.

There are some few minor issues that I think the authors should be aware of:

1. Abstract.
The same idea appears repeated twice in lines 18-20:
"This work examines the general attitude of these stu-18 dents towards vaccination. Method. General attitude towards vaccination was ex-19 amined in students attending..."

2. Lines 152-163 and tables 1 and 2.
The information regarding the sample should be relocated to the methods section. It's not a result.

3. Table 4. 
Is this table necessary? I'm not sure why it's relevant to inform about the number and % of participants showing negative scores.
Furthermore, this is a too long table that doesn't add any particularly valuable information

4. Would it be possible to report bivariate comparisons and pairwise comparisons in the same table showing mean and standard deviation values?
That is, would it be possible to join tables 3, 5 and 6?
In case the result is too large, you could build a table for beliefs, another for behavios and another for general attitude.

5. Table 6 is difficult to understand.
Why does "OR (95CI%)" appear twice in the first line?

Formal issues:
1. Keywords should be in alphabetic order.
2. References must be adapted to the journal guidelines: https://www.mdpi.com/journal/vaccines/instructions#references
3. When displaying p-values, only the part of the decimal number should be written. I mean, it is not p= 0.001, but p= .001.
4. The title of the tables must always appear in the same place, according to the rules of the journal. Currently, in some tables it is above and in other tables below the table.

Author Response

Response to Reviewer 2 Comments

Point 1: Abstract. The same idea appears repeated twice in lines 18-20: "This work examines the general attitude of these stu-18 dents towards vaccination. Method. General attitude towards vaccination was ex-19 amined in students attending..."

The same idea repeated twice is removed from the summary. Only the sentence is left “This work examines the general attitude towards vaccination in students attending the Faculty of…”

Point 2: Lines 152-163 and tables 1 and 2. The information regarding the sample should be relocated to the methods section. It's not a result.

Lines 152 to 163 refer to the number of students who answered the questionnaire, emphasising that there were at least 30 participants per course, except in podiatry where there were 28. We understand that this is part of the results and not of the methodology, as it is a data that is not known until the questionnaire is distributed. If the theoretical sample size had been calculated, it would have been included in the methodology, but as this data is obtained after the fieldwork, it should be included as a result.

In relation to table 1, the same applies to the last two columns, where reference is made to the students who responded to the questionnaire. The first column is for information and comparison. Table 2 refers only to the characteristics of the students who answered the questionnaire, for the reasons given above it cannot be included in the methodology, because it is data obtained after the questionnaire has been completed.

Point 3: Table 4. Is this table necessary? I'm not sure why it's relevant to inform about the number and % of participants showing negative scores. Furthermore, this is a too long table that doesn't add any particularly valuable information.

The entire table 4 has been modified. It is a necessary table as it allows us to know what percentage of students have negative attitudes towards the use of vaccines and how these attitudes can have a negative influence on patients. It has also been made more summarised with the most relevant information.

Point 4: Would it be possible to report bivariate comparisons and pairwise comparisons in the same table showing mean and standard deviation values?
That is, would it be possible to join tables 3, 5 and 6?
In case the result is too large, you could build a table for beliefs, another for behavios and another for general attitude.

It is not possible to join in the same table the bivariate and multivariate analysis of tables 5 and 6 because they compare different variables. Table 5 compares beliefs, behaviours and general attitude with gender, age, grade and year. Table 6 compares sex, age, grade, beliefs, behaviours and general attitude towards vaccines with influenza vaccination. In table 3 we look for differences between the means of behaviours, beliefs and general attitude with respect to sex, grade, year ..... Because of the above, behaviours, beliefs and general attitude cannot be merged in a single table or in three different tables. Moreover, the same independent variables are not considered in all tables.

Point 5: Table 6 is difficult to understand. Why does "OR (95CI%)" appear twice in the first line?

Table 6 shows a bivariate and multivariate analysis of the different variables with respect to influenza vaccination. The second, third and fourth columns show the bivariate analysis and the fifth, sixth and seventh columns show the multivariate analysis of the bivariate analysis from which significant differences were obtained, always with respect to influenza vaccination. "OR (95CI%)" appears twice in the first line because this type of analysis was carried out in both the bivariate and multivariate analyses. 

Point 6: Keywords should be in alphabetic order.

The keywords are sorted alphabetically

Point 7: References must be adapted to the journal guidelines: https://www.mdpi.com/journal/vaccines/instructions#references

All bibliographic citations are changed from superscript to number in square brackets.

Point 8: When displaying p-values, only the part of the decimal number should be written. I mean, it is not p= 0.001, but p= .001.

In all p-values found in the document, the integer part has been removed, leaving only the decimal part.

Point 9: The title of the tables must always appear in the same place, according to the rules of the journal. Currently, in some tables it is above and in other tables below the table. 

The titles of all tables are at the top.

Reviewer 3 Report

The manuscripts describes the attitude, beliefs and behavior of healthcare trainees in three different specialities towards vaccination. The authors found out that nursing students had better outcomes compared to their peers in Physiotherapy and Chiropody. The study focuses on the use of questionnaires and has a good size sample of 934 participants based on heir sample size requirement analysis. Several other studies similar to this have been reported by other investigators as indicated by the authors. Despite these previous studies which were done in other countries, this study may be beneficial in Spain. A few thing to consider to strengthen the quality of the study are:

1). Authors should suggest why they think female participants reported more positive attitude towards vaccination than males, as they indicate in line 171 and 227. Could it be because there were fewer male participants in the study? As a followup, the authors indicate that 310 respondents are needed for 95% confidence interval, since they have 191 males. if they randomly selected 191 females to have equal participants, would this change this conclusion/observations?

2) The authors state in the results line 173, that "...belief and general attitude score significantly improved year on year as the trainees progressed ...". The cited table 3 data seem to indicate that behavior and general attitude are what improved and not belief as 3rd year was the lowest for belief, could the authors clarify this?

3). Did the respondents who were hesitant or gave lower score have any reasons? Did the questionnaires include any comments sections to give an opportunity for trainees personal views?

4). While this is minor, I think that the title of the manuscript is a little misleading. The study design doesn't indicate that this study was designed to gauge trainees' attitudes towards vaccines based on anticipated vaccinations for COVID-19, neither is the study related to the pandemic. having a title that includes COVISD-19 while the study design doesn't suggest the relationship sounds misleading to me.

5). There are a few typos that can be corrected with proof reading.

Author Response

Response to Reviewer 3 Comments

Point 1. Authors should suggest why they think female participants reported more positive attitude towards vaccination than males, as they indicate in line 171 and 227. Could it be because there were fewer male participants in the study? As a followup, the authors indicate that 310 respondents are needed for 95% confidence interval, since they have 191 males. if they randomly selected 191 females to have equal participants, would this change this conclusion/observations?

During the academic year in which the study was carried out, the percentage of male students enrolled in the faculty was 23,6%. Taking into account that in the study the % of males was 20.5% we consider that they are well represented. If only 191 females had been randomly selected for the study, males would have been over-represented in the sample, not corresponding to the reality of the faculty. A paragraph was added to the discussion stating "it is not clear why women have a more positive attitude towards vaccination than men".

Point 2. The authors state in the results line 173, that "...belief and general attitude score significantly improved year on year as the trainees progressed ...". The cited table 3 data seem to indicate that behavior and general attitude are what improved and not belief as 3rd year was the lowest for belief, could the authors clarify this?

According to table 3 the mean beliefs and general attitude improve significantly in the nursing degree. In the text "in the nursing degree" is added.

Point 3. Did the respondents who were hesitant or gave lower score have any reasons? Did the questionnaires include any comments sections to give an opportunity for trainees personal views?

In relation to these questions, it was not the aim of this study to find out the reasons that led to hesitancy about vaccination and lower scores. For this reason, we did not leave an open question for them to state their reasons. This would be another possible line of research, but was not considered in the current study.

Point 4. While this is minor, I think that the title of the manuscript is a little misleading. The study design doesn't indicate that this study was designed to gauge trainees' attitudes towards vaccines based on anticipated vaccinations for COVID-19, neither is the study related to the pandemic. having a title that includes COVISD-19 while the study design doesn't suggest the relationship sounds misleading to me.

It was decided to include in the title that the work was carried out before the pandemic, because surely if it were carried out now, all the aspects studied would improve considerably. Moreover, it is an ongoing project to study what changes have occurred after the pandemic.

Reviewer 4 Report

This study reports the attitudes to vaccination (largely influenza) pre COVID-19 in a Spanish Health science undergraduate students.  This data is of historic value as it will serve to establish how attitudes have undoubtedly changed from 2021 onwards.   A detailed analysis supports the conclusions given.

Minor point: Mention which questionnaire is used in the abstract (line 21).

Author Response

Point 1: Mention which questionnaire is used in the abstract (line 21).

The name of the questionnaire used is added in the abstract.